# Social Media and Social Support: A Framework for Patient Satisfaction in Healthcare

**Md Irfanuzzaman Khan** [1,*], **Zoeb Ur Rahman** [2], **M. Abu Saleh** [1] and **Saeed Uz Zaman Khan** [1]

[1] Canberra Business School, University of Canberra Bruce Campus, Canberra 2617, Australia; abu.saleh@canberra.edu.au (M.A.S.); saeed.khan@canberra.edu.au (S.U.Z.K.)
[2] London School of Commerce, Dhaka Centre, Dhaka 1212, Bangladesh; zoeb.rahman@lsclondon.co.uk
\* Correspondence: irfan.khan@canberra.edu.au

**Abstract:** Social media has been a powerful source of social support for health consumers. In the healthcare sector, social media has thrived, building on various dynamic platforms supporting the connection between social relationships, health, and wellbeing. While prior research has shown that social support exerts a positive impact on health outcomes, there is scant literature examining the implications of social support for patient satisfaction, which suggests that there is a profound gap in the extant literature. The objective of this study is to develop and test a theoretical model for understanding the relationship between different dimensions of social support and patient empowerment. The study further investigates the debated relationship between patient empowerment and patient satisfaction. The measurement model indicated an acceptable fit ($\chi^2$ = 260.226; *df*, 107, $\chi^2/df$ = 2.432, RMSEA = 0.07, GFI = 0.90, IFI = 0.95, TLI = 0.94, and CFI = 0.95). Findings indicate that emotional support ($p < 0.001$), information support ($p < 0.05$), and network support ($p < 0.001$) positively influence the notion of patient empowerment. In turn, patient empowerment positively influences patient satisfaction ($p < 0.001$). The proposed framework contributes to the health communication literature by introducing a novel framework for patient satisfaction in the social media context, which provides important inputs for healthcare service providers in developing patient empowerment strategies.

**Keywords:** social media; social support; patient satisfaction; online healthcare; patient empowerment; patient engagement

## 1. Introduction

The application and use of social media in healthcare has grown rapidly. Consequently, healthcare providers are employing social media platforms to provide social support in order to empower patients while improving health outcomes [1,2]. Since its emergence in the early 2000s, social media is now a powerful source of social support for patients. In the healthcare sector, social media facilitates the connection between social relationships, wellbeing and health [3,4]. Virtual communities utilise social media not only to access information about their medical condition but also, more importantly, to share with others the daily emotional aspects of one's life [5]. Consequently, social media is being utilised by healthcare providers to harness social support, thus improving health outcomes by meeting basic human needs for companionship, intimacy, a sense of belonging, and reassurance of one's worth as a person [6]. Today, many healthcare organisations, patients and community support groups provide numerous social support strategies for patients [7,8].

Contemporary discussion in the healthcare domain already established that social media can offer a vast amount of health-related information to which multiple stakeholders contribute, including healthcare professionals (i.e., doctors) and lay people (i.e., patients) [9]. Health communities around the world are using various social media platforms to involve patients in their discussions and interactions in order to improve their self-care and health outcomes [10]. The existing literature has revealed that patient participation in health-related discussions on social media and other online forums results in

improved communication, information exchange between patients and with healthcare professionals [11]. Moreover, a recent study revealed that social support results in well-being and happiness in the digitalised healthcare environment [11]. All of these examples point towards the fact that social media now sits at the centre of the social support exchange between patients on various social media platforms.

Although some prior studies have recognised many benefits of using social media as a means of social support [10,12], not much research has addressed the relationship between social support, patient empowerment, and patient satisfaction in the healthcare context. While health communities continue to participate in social media-based health communities for information seeking and emotional support, very little is known regarding the types of support offered in online platforms and whether the dimensions of online support can influence patient empowerment outcomes [13]. This research is an attempt to examine whether different types of social support shape patient empowerment outcomes and patient satisfaction. Generally speaking, there is scant literature examining the impacts of various forms of social media based on social support concerning patient empowerment and patient satisfaction. For this reason, the primary objective of this study is to propose a research model that seeks to explain the impacts of social media-based support on patient empowerment and patient satisfaction. In the following sections, firstly, we review and contextualise the social support literature; secondly, we explain the different dimensions of social support to conceptualise the antecedents of patient empowerment and satisfaction. Thirdly and lastly, the research methods and results are presented. The study concludes with the discussion, implications and limitations and conclusion.

## 2. Literature Review

This literature review section sheds light on the relationship between social support, patient empowerment and patient satisfaction.

### 2.1. Social Support

Social support is a philosophy which is envisioned to facilitate belongingness, coping, esteem, and competence through actual or perceived exchanges of psychosocial resources [14]. Many studies argue that social support is a multidimensional construct consisting of various forms of support, including informational support, emotional support, instrumental support, and companionship. Social support has been demonstrated as exerting an impact on various health outcomes such as psychological wellbeing, emotional support, and companionship [15].

Furthermore, social media facilitates social storytelling, which is another important characteristic of social media-based communities [16,17]. For example, using Facebook and Twitter to share narratives about various diseases is a common method used by healthcare service providers. Digital narratives have been successfully applied in interventions designed to raise awareness and to improve screening rates in breast, colorectal, and prostate cancers [18,19]. The power of storytelling is often used to provide social support in the healthcare scenario. However, the skilled facilitation of storytelling is extremely important here and, for example, Fiddian-Green et al. [20] suggested some important considerations. There are important ethical aspects concerning the application of online storytelling involving vulnerable populations. The authors recommended delving deeply into ethical issues including, but not limited to the ownership and sharing of digital materials, the reduction of harm in a group context, particularly with regard to potentially uncooperative group dynamics or the disclosure of trauma; and the potential reproduction of stigmatising narratives [20].

Smailhodzic and Hooijsma [6] explained four categories of social support in the context of social media in healthcare. These are emotional support, esteem support, information support, and network support. The above categories of social support are summarised below in Table 1.

**Table 1.** Categories of social support.

| Categories | Features | Examples |
|---|---|---|
| Emotional Support | Primarily, the individuals' emotional and/or affective needs are attended to by providing emotional support [21]. In the health context, support groups focus on providing a responsive environment in which patients' painful emotions can be felt and understood by others as perceived by the patient and thus minimise feelings of isolation [22]. | PatientsLikeMe (http://www.patientslikeme.com, accessed on 27 August 2021) is an example of providing patients with emotional support since it: aggregates isolated fragments of health information and facilitates the creating and sharing of community resources through storytelling [23]. |
| Esteem support | Esteem support relates to exhibiting encouragement and hope for individuals to manage a particular health condition [24]. | Cancer patients rely on social media to discuss their illness experiences, seek advice and learn from and support each other [25]. |
| Information support | Information support is predominantly perceived as the most popular social support activity that is built on the philosophy of providing patients with useful and relevant information according to their health conditions, treatment options and other health-related issues [21,26]. | Websites such as WebMD (http://www.webmd.com, accessed on 15 June 2021) and MedHelp (http://www.medhelp.org, accessed on 15 June 2021) provide individuals seeking health information related to disease assessments, health/treatment advice, sharing experiences and question-answer sessions in closed or open groups. Medical blogs written by healthcare professionals are also published on these sites. |
| Network Support | Network support enables patients with a specific health disorder to connect with other patients experiencing similar conditions and provides them with a sense of belonging [27]. | Online communities related to breast cancer provide relevant information on treatment, managing symptoms, prevention advice, and emotional support for the patients [28]. |

Although online social support groups offer significant benefits for healthcare communities, there are some inherent limitations associated with social media-based support groups. For example, a systematic review of 378 citations reporting clinical outcomes on leading contemporary social media channels revealed that the overall impact of social media-based information exchanged on chronic disease was variable, with 48% of studies indicating benefit, 45% neutral or undefined, and 7% suggesting harm [29]. Other studies also indicated that the impact of social media-based support is inconsistent, and offers limited engagement despite being low cost [30]. For example, one recent study shows that emotional communication competence moderates the effects of providing and receiving emotional support, and individuals with lower emotional communication competence tend to experience detrimental impacts on emotional well-being compared to those with higher emotional competence [31]. Another area that has come under increased scrutiny is the quality of health-related information and propagation of misinformation in online patient support groups. In health-related social media channels the quality of information appears to be poor, which can be detrimental to patients' health and wellbeing [32].

However, a majority of these studies can confirm that social media platforms are playing a key role in providing valuable social support in difficult times, and during important health events [30,33]. In their study, Barak, Boniel-Nission, and Suler [34] observed that empirical research frequently outlines limited or no specific outcomes for online support groups. Nonetheless, it points towards non-specific individual impacts such as psychological wellbeing and personal empowerment, which is a much needed force while battling a specific health condition. The notion of patient empowerment is discussed in more detail below.

*2.2. Patient Empowerment*

Before the emergence of social media, patients had limited options in regard to sharing experiences with a healthcare provider. Provider reputation and peer recommendations played an influential part in choosing a medical professional or hospitals/clinics. Subsequently, accessing information concerning the quality of a specific care provider was

complicated in the pre-social media era. At the present time, social media channels are increasingly used by patients to engage, share and rate their encounters with healthcare service providers [35,36]. They are leveraging social media technologies to connect with patients experiencing the same or a similar illness and how to manage it [37]. This interactive engagement process results in patient empowerment, which is based on the notion of enhanced self-awareness, skills, and the knowledge required to achieve health-related goals [6]. According to Oh and Lee, [38] information support, esteem support, and emotional support provided by various social media-based patient groups act as an important antecedent of a patient's sense of empowerment. The three subcategories of empowerment documented by Smailhodzic and Hooijsma [6] are improved subjective well-being, heightened psychological well-being, and improved self-management and control. A summary of these categories is presented in Table 2.

**Table 2.** Three subcategories of empowerment.

| Category | Comments |
| --- | --- |
| Subjective well-being | Subjective well-being is the patient's state of emotional satisfaction resulting from their use of social media for health-related reasons. |
| Psychological well-being | Developing and experiencing a positive relationship with other patients through social media usage. |
| Self-management and control | The patient's ability to manage their health condition is improved. It includes better coping strategies, effective self-management, and an enhanced ability to manage the illness, its related adverse health conditions and support of chronic disease self-management. |

The evidence concerning the effectiveness of these online support channels on an individual's health is not circumstantial as it has been documented that both health professionals and patients are effectively using these channels and showcasing the benefits of participation in social media-based communities. However, the downside of social media driven empowerment should also be discussed. Increased interactions and involvement in the clinical decision-making process means that healthcare professionals may not be in complete control of decisions being made during clinical interactions; this may increase the risk for healthcare professionals [6,39]. In fact, some medical experts perceive online support groups as a risk to their professional expertise, and control over clinical interactions which acts as a significant deterrent to harvesting the benefits offered by social media based support groups [39]. Nevertheless, the notion of patient empowerment continues to appear in medical debates, and it will be interesting to observe the impacts of empowerment on patient satisfaction.

*2.3. Patient Satisfaction*

Patient satisfaction is a complex domain and there is no global consensus on the determinants of patient satisfaction. However, literature demonstrates that social media applications enable healthcare professionals to engage with their patients, increase brand image and patient satisfaction in a very cost-effective manner [40–42]. A recent study conducted on 390 hospitals revealed that there is a positive association between hospital Facebook activity and patient satisfaction. It emerged that hospitals that had a Facebook page were active on Facebook in the past 30 days, and had more "likes", had more patients willing to definitely recommend the hospital, and had a higher overall satisfaction score [43].

From a holistic perspective, patient satisfaction encompasses various aspects of the healthcare system, including empathy, level of care, physical environment, and caregiver characteristics [44,45]. The extant literature primarily elaborated on the relationship between patient satisfaction and the quality of healthcare [44,45]. Based on the above analysis, this study presents a different perspective on the concept of patient satisfaction, and examines patient satisfaction from the angle of social support-driven empowerment.

### 3. Research Model

The proposed research model seeks to examine the relationships between four distinct dimensions of social support and patient empowerment in the social media context. The study further investigates the relationship between patient empowerment and patient satisfaction in the social media context. Consequently, this study contributes to the body of knowledge in social media-based healthcare by offering a fresh take on the key dimensions of online social support. Previously, researchers demonstrated a positive link between online social support and empowerment, self-efficacy, knowledge, and health-promoting behaviours [46]. However, the individual effects of esteem support, emotion support, information support and network support are relatively unknown. In order to address these inherent limitations in the extant literature, five hypotheses were devised for the proposed research model provided in Figure 1.

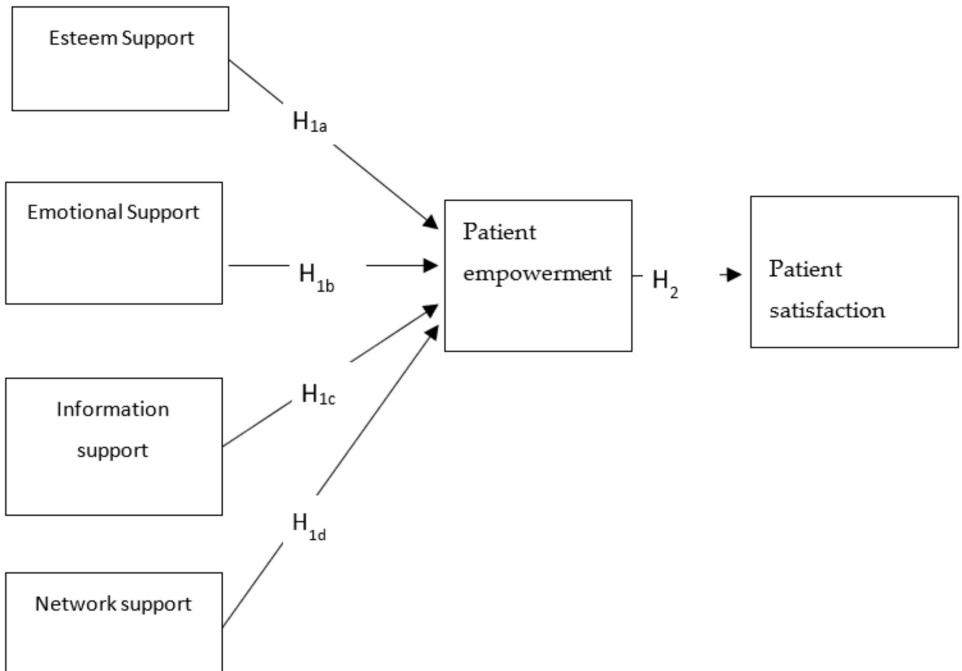

**Figure 1.** Research Model.

It has been well established in contemporary literature that there is a positive association between social support and patient empowerment. This research proposes four separate dimensions of social support; namely, esteem support, emotional support, information support and network support, which exert a distinct impact on the notion of patient empowerment in the online healthcare context. In healthcare, esteem support is instrumental in encouraging patients to take actions to manage certain illness, and getting support patients' encouragement [25]. Thus, self-esteem can be considered a driver of patient empowerment. On the other hand, social media can also act as an avenue for emotional support. Social media enables communication and interaction between individuals who are experiencing similar health conditions [2]. Consequently, emotional support can act as catalyst of patient empowerment.

The third dimension, Information support, can also be considered as an important driver of patient empowerment. Information support provided through social media channels are diverse and can range from symptom management, health tips, self-diagnosis routines, and wellbeing [47,48]. Finally, network support is another important dimension of social media-based social support in healthcare. These social media-based health communities enable patients to connect and develop healthy relationships and bonds without being near to other participants [24]. As a consequence, social media can augment social connections among patients by overcoming the communication barriers in traditional

healthcare [49]. For example, a recent study concerning the role of online social support in the social and well-being of unwed single mothers in China reveals that online social support results in improving the wellbeing of single mothers [50].

Consequently, many healthcare service providers have already created a dynamic, interactive online presence to encourage community engagement, and this has been applied in clinical practice, education, research, and administration. Many hospitals use social media as a tool for marketing to and involving their patient base. Physician practices are also using Facebook pages, Twitter accounts, and blogs in an effort to expand their practice and earn patient referrals [51,52]. According to Oh and Lee [38], information support, esteem support, and emotional support provided by various social media-based patient groups act as an important antecedent of a patient's sense of empowerment. Based on the four dimensions of social support and their effects on the notion of patient empowerment, the following hypotheses have been proposed.

**Hypothesis 1a.** *Esteem support promotes patient empowerment.*

**Hypothesis 1b.** *Emotional support facilitates patient empowerment.*

**Hypothesis 1c.** *Information supports enable patient empowerment.*

**Hypothesis 1d.** *Network support positively influences patient empowerment.*

The proposed research also looked at the relationships between patient empowerment and patient satisfaction. In healthcare, patient empowerment enriches the level of satisfaction [53]. Similarly, Yeh and Wu [54] demonstrated that empowerment improves patient health, and there is a positive relationship between patient empowerment and patient satisfaction. In an online health communication context, social media plays a pivotal role in ensuring patient engagement and satisfaction [55]. Thus, the following proposition is considered:

**Hypothesis 2.** *There is a significant and positive relationship between patient empowerment and satisfaction in the social media context.*

## 4. Method

This is a cross-sectional study, and the data were collected from individuals who are members of online healthcare groups in Bangladesh. This involved a field survey of online healthcare group participants. A quantitative survey is considered to be an appropriate data collection method for this research, as research conducted in a similar domain pursued quantitative surveys to test the impacts of independent variables on dependent variables [56,57]. The survey was administered to 629 individuals at several tertiary education institutions in Dhaka. The primary reason why university students, faculty members and administrators were selected as the sample is that the above-mentioned group are likely to be frequent users of social media and the internet in Bangladesh. To correctly identify the user of online healthcare groups, a screening question was used: "Did you visit any one of the online healthcare advice or support group in the last year?" Respondents who replied 'yes' were invited to participate and complete the survey. In total, 234 respondents completed questionnaires and the response rate amounted to 26.88%.

### 4.1. Participant Characteristics

Of the 234 individuals who participated in this survey, 56% were male and 44% were female. The age of the respondents varied widely; most participants were in the 18–35 (54%) cohort. Additionally, 24% of the respondents were between 35–49 years of age, while 14% were between the ages of 50 and 59, and the remaining 8% were 60 years and older. In terms of education, 40% completed HSC, 38% completed undergraduate degrees, 9% completed the Master's degree, and 4% completed a PhD degree. The remaining 9%

acquired vocational qualifications. In terms of the frequency of using social media, 76% are frequent users. A breakdown of the participant characteristics is provided in Table 3.

**Table 3.** Demographic information.

| Category | | Frequency | Percent |
|---|---|---|---|
| Gender | Male | 132 | 56% |
| | Female | 102 | 44% |
| Age | 18–35 | 126 | 54% |
| | 35–49 | 56 | 24% |
| | 50–59 | 33 | 14% |
| | 60 years and above | 19 | 8% |
| Education | HSC | 93 | 40% |
| | Undergraduate | 88 | 38% |
| | Masters | 21 | 9% |
| | PhD | 9 | 4% |
| | Others | 23 | 9% |
| Social media usage frequency | Frequently | 178 | 76% |
| | Occasionally | 46 | 20% |
| | Rarely | 10 | 4% |

*4.2. Questionnaire Items*

The questionnaire items were sourced from various measures, with some minor modifications to fit the social media context of this study. The measures for emotional support and esteem support were adopted from the work by Deng and Liu [58]. Measures for information support were adopted from the work of Liang and Xue [59]. Meanwhile, the measures for network support were contextualised from the article by Haslam and Tee [60]. Patient empowerment measures were borrowed from Johnston and Worrell [13]. Finally, indicators for patient satisfaction were developed from Wu and Deng's analysis [61]. The full-scale, factor loadings, and reliability scores are provided in Table 4 below. Each questionnaire item was weighed on the basis of its factor loading for its corresponding constructs, and each construct illustrates high internal consistency and reliability, as demonstrated by the Cronbach's alpha score reported in Table 4.

*4.3. Quantitative Data Analysis*

After ensuring construct reliability, confirmatory factor analysis (CFA) serves to examine the validity of the proposed research theoretical framework, and the underlying dimensions of the constructs [62]. Following CFA analysis, the structural equation modelling (SEM) technique was employed to examine the hypothesised relationships. SEM is typically used in the field of social sciences to examine linear causal relationships among variables [63]. SEM represents the observed data through structural parameters defined by a theoretical framework [64,65]. Moreover, SEM is also well known for its ability to account for measurement error, which is a common limitation observed in many studies [66]. Considering causal relationships identified in the proposed research model (Figure 1) of this study, it was logical to use SEM to examine the relationship between the constructs articulated in this paper.

*4.4. Measurement Model Analysis and Construct Validity*

Confirmatory factor analysis was carried out after incorporating all variables. The result revealed a satisfactory model fit CMIN = 207.634 with df = 103 and CMIN/df ratio = 2.016; IFI = 0.97; TLI = 0.96; CFI = 0.97; and RMSEA = 0.06. Therefore, overall fit was at an acceptable level. An assessment of the factor loading ensured all retained measurement items exceeded the recommended universal cut-off factor loading of 0.50 [65].

**Table 4.** Construct measures.

| Construct | Indicators | Factor Loadings | Composite Reliability | Cronbach's Alpha | AVE |
|---|---|---|---|---|---|
| Esteem Support | Show confidence in my ability to address my health concerns. | 0.75 | 0.86 | 0.89 | 0.68 |
| | Makes me feel that I am good at making healthy decisions. | 0.90 | | | |
| | Makes me feel as though I can handle my health issues. | 0.82 | | | |
| Emotional Support | Provides me with encouragement. | 0.87 | 0.91 | 0.94 | 0.77 |
| | Makes me feel well and good. | 0.90 | | | |
| | Makes me feel relaxed. | 0.87 | | | |
| Information Support | Follow the advice offered by online health information. | 0.71 | 0.82 | 0.83 | 0.60 |
| | Online health information heavily influences my health-related decisions. | 0.78 | | | |
| | I use online health information to cope with my emotions, such as fear, stress, and frustration. | 0.83 | | | |
| Network Support | I rely on support from friends and others online. | 0.93 | 0.95 | 0.92 | 0.86 |
| | When facing difficult situations, I am likely to discuss them with online networks. | 0.92 | | | |
| | I receive support from my online community. | 0.93 | | | |
| Patient Empowerment | I am confident about my ability to manage my personal healthcare needs. | 0.90 | 0.89 | 0.92 | 0.74 |
| | I have significant autonomy in determining how I manage my personal healthcare. | 0.88 | | | |
| | I can make my own decisions regarding managing my personal healthcare. | 0.79 | | | |
| Patient Satisfaction | Interacting with doctors on social media makes me feel satisfied. | 0.93 | 0.94 | 0.95 | 0.83 |
| | Interacting with doctors on social media makes me feel happy. | 0.92 | | | |
| | Interacting with doctors on social media makes me feel content. | 0.89 | | | |

Construct reliability values were above the minimum acceptable benchmark of 0.70 [67]. Discriminant validity was addressed by comparing the average variance extracted (AVE) values with the squared correlation between each pair of constructs [68]. The AVE for each construct exceeded the inter-construct correlations, confirming the discriminant validity. The AVE for each latent construct is above the 0.50 threshold, ranging from 0.63 to 0.85, demonstrating an adequate depiction of convergent validity. The correlation matrix of latent variables is shown in Table 5 below.

**Table 5.** Correlation matrix.

| | 1 | 2 | 3 | 4 | 5 | 6 |
|---|---|---|---|---|---|---|
| Esteem Support | **0.824** | | | | | |
| Emotional Support | 0.429 ** | **0.877** | | | | |
| Information Support | 0.399 ** | 0.456 ** | **0.774** | | | |
| Network Support | 0.378 ** | 0.552 ** | 0.370 ** | **0.927** | | |
| Patient Empowerment | 0.595 ** | 0.451 ** | 0.393 ** | 0.465 ** | **0.860** | |
| Patient Satisfaction | 0.511 ** | 0.452 * | 0.348 ** | 0.467 * | 0.681 | **0.911** |

Notes: Correlation is significant at the 0.05 * and 0.01 ** levels. The bold face values indicate the square root of AVE.

## 5. Results

This study used the structural equation modelling (SEM) technique to test the hypotheses developed for this study. SEM is a collection of statistical techniques that allow researchers to examine hypothesised relationships between one or more independent variables (IVs), either continuous or discrete, and one or more dependent variables (DVs) [69]. The assessment of the model fit for the measurement revealed an acceptable model fit based on the recommendations of Beran and Violato [65]. The $\chi^2/df$ is 2.43 and lower than 3. RMSEA is 0.07 and lower than 0.08. The other indices (i.e., CFI, IFI and TLI) are all above 0.90 ($\chi^2$ = 260.226; *df*, 107, $\chi^2/df$ = 2.432, RMSEA = 0.07, GFI = 0.90, IFI = 0.95, TLI = 0.94, and CFI = 0.95). Subsequently, it can be stated here that the measurement model exhibits an acceptable model fit. Estimates derived from the SEM analysis were

used to test the research hypotheses. The results of hypothesis testing are provided in Table 6 and they generate support for all hypothesised relationships depicted except H1a in the theoretical model. The results strongly suggest that emotional support, information support and network support significantly and positively influence the notion of patient empowerment. Therefore, hypotheses 1b, 1c and 1d are all found to be true. Surprisingly, hypothesis 1a was not supported. For the impact of patient empowerment on patient satisfaction, the results suggest that patient empowerment significantly influences patient satisfaction. Thus, hypothesis 2 is found to be true. These outcomes demonstrated that enhanced social support through social media channels for patients and health consumers does result in increased patient empowerment. Consequently, patient empowerment is a strong determinant of satisfaction in the healthcare context.

**Table 6.** Hypothesis Testing.

| Path | | | Estimate | S.E. | C.R. | *p* | Result |
|---|---|---|---|---|---|---|---|
| Esteem Support | → | Patient Empowerment | 0.110 | 0.071 | 1.544 | 0.122 | Not supported |
| Emotional Support | → | Patient Empowerment | 0.347 | 0.113 | 3.062 | *** | Supported |
| Information Support | → | Patient Empowerment | 0.161 | 0.071 | 1.985 | 0.043 | Supported |
| Network Support | → | Patient Empowerment | 0.311 | 0.073 | 4.243 | *** | Supported |
| Patient Empowerment | → | Patient Satisfaction | 0.640 | 0.070 | 9.113 | *** | Supported |

*** Significant at the 0.001 level.

## 6. Discussion

This study aims to investigate the antecedents of patient satisfaction. Firstly, this article tested the causal effect of four social support dimensions on patient empowerment. Second, this research tested the influence of patient empowerment on patient satisfaction. Consequently, this research endeavour provides valuable insights for developing health communication and online support strategies. Several findings are explained more fully below.

In H1a, the finding exhibited an insignificant relationship between esteem support and patient empowerment ($\beta = 0.110$, $p > 0.05$). Prior studies have studied the effects of esteem support from different perspectives and reported positive results. Valkenburg and Peter [70] reported that social self-esteem positively influences an individual's wellbeing. In contrast, Deng and Liu [58] found that self-esteem significantly influences health self-efficacy. Thus, the result contradicts the earlier findings and does not provide support for the notion that a significant relationship exists between esteem support and patient empowerment. This is an interesting finding and needs further investigation within a clinical setting. For example, recent research on 34 cancer patients demonstrates that learning from fellow cancer patients' stories online has a significantly greater influence on cancer patients' perceptions and expectations compared to medical professionals' influence [25].

In H1b, this research demonstrates that emotional support significantly influences the notion of patient empowerment ($\beta = 0.347$, $p < 0.001$). The finding is consistent with previous studies which also illustrate how online emotional support plays an important role in empowering patients to improve their health outcomes [71,72]. A recent multi-method study infers that social support and information utility derived from online health community participation helps to shape perceptions of patient empowerment among community participants [13]. Thus, it is apparent that more investment is required to facilitate emotional support which empowers patients and produces more patient empowerment.

Referring to H1c, the positive relationship between information support and patient empowerment signifies that interacting with others in social media-based support groups enhances their ability to take more informed medical decisions ($\beta = 0.161$, $p < 0.005$) [73]. However, more caution needs to be observed in terms of facilitating information support. A recent study on colorectal cancer patient survivors demonstrated that the need for in-

formational support changes over time, and individual patient characteristics play a vital role in terms of information requirements [74]. Similarly, this study detected a positive relationship between network support and patient empowerment ($\beta = 0.311$, $p < 0.001$) (H1d). Hence, network support can directly influence health consumers' sense of confidence and efficacy while interacting with medical professionals and managing their health conditions. The benefits of network support have been documented in other research [24,75]. According to Naslund and Aschbrenner [24], patients with serious mental illness report significant benefits from various forms of network supports, such as online peer interaction or story sharing.

Lastly, this study also demonstrates a positive relationship between patient empowerment and patient satisfaction ($\beta = 0.640$, $p < 0.001$) (H2). However, it must be noted that empowerment is a multi-dimensional construct, and social support is an important but not the sole dimension of patient empowerment. Prior studies indicated that critical characteristics of patient empowerment include social support, patient education, disease awareness and self-efficacy [54]. Although other research has shown that patient empowerment is connected to healthcare quality [76], the relationship between patient satisfaction and patient empowerment is still not well understood. For example, Tartaglione and Cavacece [77] found no association between patient empowerment and patient satisfaction. With this in mind, this study provides new insights that in developing countries like Bangladesh, patient empowerment will play a significant role in patient satisfaction.

*Implications and Limitations*

Our contribution to the literature lies in the development of a novel theoretical framework presented herein which considers different domains of social support, namely emotional, esteem, information, and network support, to enhance the current understandings of the determinants of patient empowerment, and satisfaction in social media-based health communities. This extends the current understanding of social support dimensions in online healthcare communities. This cross-sectional study reflects on the relationship of the four dimensions of social support [6], patient empowerment and patient satisfaction. Firstly, we empirically tested the significant effects of the four dimensions of social support on patient empowerment, as well as the relationship between patient empowerment and patient satisfaction. These findings enrich the research on the dimensions of social support and patient satisfaction.

Secondly, this study examined the relationship between social support dimensions, patient empowerment and patient satisfaction in a developing country context. The results of this research confirm the need for providing online social support, which is likely to empower patients. The dimensions of online social support are often overlooked by healthcare providers in a developing country such as Bangladesh. The findings will also prompt healthcare providers to undertake initiatives that facilitate social media-based social support for healthcare consumers. An in-depth understanding of different dimensions of social support can help healthcare providers design and offer more targeted online services for patients and customers.

Finally, this research endeavour expands our knowledge of the social support dimensions of social media networks. The quantitative analysis performed for testing the research hypotheses also offers validation for the necessity of inspiring and supporting the use of social media to improve the health outcomes of healthcare consumers.

This study is not free from certain limitations. Firstly, this is a cross-sectional study due to time and resource constraints. Secondly, the sample chosen for this study comprises internet and social media users. Therefore, the results do not capture the perspectives or opinions of non-social media users. Thirdly, this study is based on a convenience sampling method and does not focus on any exclusive healthcare consumer group. Future research can be conducted on a specific clinical setting (for example, social support for people with an eating disorder) and social media platforms. Fourthly, this study only considers the active users of social media and their interactions with social support dimensions. It must

be acknowledged that many patients who do not use social media for health-related matters may also feel empowered, supported or satisfied for other reasons, such as knowledge, access to infrastructure or having skills in self-efficacy. Thus, future studies should also consider the notion of bidirectionality. Further research should also examine whether the patients who feel more empowered, satisfied and need social support are more active in the healthcare domain of social media. Fifth, there are other micro and macro factors which play an important role in shaping how patient satisfaction is shaped and perceived. Finally, moderating factors such as trust and facilitating conditions should be incorporated to further validate the findings.

## 7. Conclusions

This paper investigates how esteem support, information support, emotional support and network support offered by social networks can improve healthcare consumers' sense of empowerment, and whether empowerment leads to better satisfaction within the healthcare social media context. These results demonstrate the importance of emotional support, information support and network support in enhancing health consumers' sense of empowerment. Given the widespread use of social media, and positive associations between social support dimensions, patient empowerment and patient satisfaction, this can lead to an optimistic proposition that social media has the potential to generate reliable online social support for patients. More significantly, this in turn is likely to have positive ramifications for an individual's health outcomes.

**Author Contributions:** Conceptualization, Z.U.R.; Formal analysis, M.A.S.; Investigation and data collection, Z.U.R. and S.U.Z.K.; Methodology, Z.U.R.; Resources, Z.U.R.; Writing—original draft, M.I.K.; Writing—review & editing, M.I.K., S.U.Z.K. and M.A.S. All authors have read and agreed to the published version of the manuscript.

**Funding:** This research received no external funding.

**Institutional Review Board Statement:** The study was approved by the London School of Commerce, Dhaka Centre, Bangladesh.

**Informed Consent Statement:** Informed consent was obtained from all subjects involved in the study.

**Data Availability Statement:** The data presented in this study are available on request from the corresponding author.

**Conflicts of Interest:** The authors declare that they have no conflicts of interest.

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
