# Peer review of "Social Media and Social Support: A Framework for Patient Satisfaction in Healthcare"

_informatics, doi:10.3390/informatics9010022_

Round 1
Reviewer 1 Report
The paper is devoted to the development of the concept of understanding the relationship between various aspects of social support and patient empowerment. The relevance of the study is dictated by the fact that social networks flourish in the healthcare sector, based on various dynamic platforms that support the connection between social relations, health, and well-being. Analysis of the current state shows that social support has a positive effect on health outcomes. However, little research has been devoted to investigating the impact of social support on patient satisfaction, indicating a deep gap in the existing literature. The authors offer a conceptual framework for understanding the relationship between various aspects of social support and patient empowerment. The discussed relationship between patient empowerment and patient satisfaction is also explored. The proposed framework offers a solution for healthcare communication by introducing a new patient satisfaction framework in the context of social media that provides important input for healthcare providers in the development of patient empowerment strategies.
Despite the satisfactory quality of the article, there are some shortcomings that need to be corrected.
- The abstract should include not only an introduction and methods but also results.
- The formatting of the link to sources is different in text, e.g. line 137.
- It is not clear from the text what the proposed concept is. The authors formulated 5 hypotheses and conducted a survey. More attention should be paid to the concept.
- The scientific novelty of the paper should be highlighted.
- The Conclusion section should be included.
- The discussion section should also discuss the obtained numerical results.
- The discussion section should compare obtained results with other research.
In summarizing my comments I recommend that the manuscript is accepted after major revision, including focusing on the model and obtained numerical results.
Author Response
|
Reviewer Comments |
Response |
|
Despite the satisfactory quality of the article, there are some shortcomings that need to be corrected.
In summarizing my comments I recommend that the manuscript is accepted after major revision, including focusing on the model and obtained numerical results.
|
1. Thank your four valuable comments. We have included the findings in abstract now. 2. Fixed 3. Thank you for your suggestions. Now, we have elaborated the concept in detail in the introduction section and also in the method section (Highlighted in blue). 4. Thank you. We have discussed the contribution of the paper in the implication section highlighted in blue. 5. Thanks for the suggestion. We have included a conclusion. 6. Thank you. We have now included the numerical results. 7. Thank you. We have aligned the findings with extant research. Highlighted in blue in the discussion section. |

Reviewer 2 Report
This study aims to develop a conceptual framework for understanding the relationship between different dimensions of social support and patient empowerment. According to the authors, the proposed framework contributes to the health communication literature by introducing a novel framework for patient satisfaction in the social media context. Thus, it provides important inputs for healthcare service providers in developing patient empowerment strategies.
This study is not well-researched and well-described, nor are the study objectives developed from the extant literature.
Nowadays, science is increasingly criticized for its lack of reliability, transparency, and repeatability. This study is well-suited to these three critics. Due to the lack of a proper review of the literature in accordance with established research study principles, this study receives no support from previous work in this field. The methodology used to conduct this study is devoid of both description and expressiveness. Numerous details about the methods are omitted or are not expressed fully. The survey instrument has not been provided. Statistical tests are not justified. The study participants' demographic information is not provided. The term 'conceptual framework' is used for what purpose? How is this a conceptual framework? Because such studies are difficult to replicate, they are ineffective for the reader. Additionally, it is unclear (a) what literature discusses this issue, (b) whether the gaps identified in this study are justified by previous studies or not (as the methodology for conducting previous studies is not detailed in this study), and (c) whether the survey data used by the authors can be relied upon or not, and how the approach can be compared to previously researched approaches. As a result, it is impossible to determine the reliability of the results reported in the paper.
The paper is riddled with grammatical and linguistic errors. A comprehensive review of proper English language usage is recommended.
Author Response
|
Reviewer’s Comments |
Changes |
|
This study is not well-researched and well-described, nor are the study objectives developed from the extant literature. Nowadays, science is increasingly criticized for its lack of reliability, transparency, and repeatability. This study is well-suited to these three critics. Due to the lack of a proper review of the literature in accordance with established research study principles, this study receives no support from previous work in this field.
.. Additionally, it is unclear (a) what literature discusses this issue, (b) whether the gaps identified in this study are justified by previous studies or not (as the methodology for conducting previous studies is not detailed in this study), and (c) whether the survey data used by the authors can be relied upon or not, and how the approach can be compared to previously researched approaches. As a result, it is impossible to determine the reliability of the results reported in the paper. The paper is riddled with grammatical and linguistic errors. A comprehensive review of proper English language usage is recommended
|
Thanks for your remarks. We have made a lot change through out the paper and attempted established a strong linkage with the extant literature. The changes made in the paper are highlighted in blue/red. We have conducted additional literature review and expanded the introduction/literature review and method section accordingly |
|
The methodology used to conduct this study is devoid of both description and expressiveness. Numerous details about the methods are omitted or are not expressed fully. The survey instrument has not been provided. |
We have provided the questionnaire items used for this study in Table -4 Construct measure- Page 11 & 12 |
|
Statistical tests are not justified. |
Thank your for your valuable input. We have added the justification under the heading quantitative data analysis section within the method. |
|
The study participants' demographic information is not provided. |
Thank you. We have now included participant’s demographic information in tables and the explanations are highlighted in blue |
|
The term 'conceptual framework' is used for what purpose? How is this a conceptual framework? Because such studies are difficult to replicate, they are ineffective for the reader. Additionally, it is unclear (a) what literature discusses this issue, (b) whether the gaps identified in this study are justified by previous studies or not (as the methodology for conducting previous studies is not detailed in this study), and (c) |
Thanks for your suggestion. We have replaced the term conceptual model with ‘Research model’ Additionally, we have included a research in the Introduction section and also expanded the literature review section and pointed towards the research gap.
We have also highlighted the methods used by previous studies in the opening paragraph of the method section. Page 10 highlighted in blue.
|
|
whether the survey data used by the authors can be relied upon or not, and how the approach can be compared to previously researched approaches. As a result, it is impossible to determine the reliability of the results reported in the paper |
Thank you for your suggestion. We have conducted extensive reliability and validity analysis highlighted in blue page 11, 12 & 13. |

Reviewer 3 Report
The paper is interesting and original. I have some comments:
- The paper is very one-sided. It examines patient empowerment and satisfaction only as a positive feature. The downsides should also be mentioned. Please see the comments and suggestions below.
- There is no control group. Might patients who do not use social media for health issues also feel empowered, supported or satisfied for other reasons? Two phenomena that occur together are not proof of causality. Bidirectionality should also be had in mind. Perhaps patients who feel more empowered, satisfied and need social support would be active on social media and be willing to participate in a study. Empowerment, satisfaction and support may come from other sources, but those patients would not have been included in your study.
Concerning my comment 1:
- Literature review, point 2.1: Do you mean that assessing information concerning a specific care provider is less complicated now? Please mention misinterpretations and faulty complaints and how claims may damage a caregiver's reputation without objective grounds.
Storytelling can be emotionally charged and also present a more exaggerated version of reality.
Being a member of a preferred group that shares your personal interests and does not allow for other views may be dangerous.
Patient empowerment may have its pitfalls. E.g. if a patient follows other patients' advice instead of the physician's instructions. The patient may feel better emotionally but may become more ill when not following the doctor's orders.
- Activity and likes on facebook (now Meta) do not necessarily mean the health care provider/hospital/clinic is the best choice for a patient. Making an appointment there may give a patient a placebo boost without necessarily meaning that the best help will be given. It is no more than free advertisement for the establishment. Having a blog and being very active in social media does not mean that the physician's practice is better than those that are less active. It only means that someone at the practice has enough time to blog and be online more than others. Is choosing the care provider with the most active blog always the best choice for a patient?
- Discuss also the downsides of patient empowerment. I think the most dangerous side is patients having their own ideas on how to treat a disease and saying 'no' to efficient treatments, and 'yes' to less efficient because of social media influence.
- For information support: Please also mention misinformation and misunderstanding information.
- Empowerment and satisfaction: If a patient asks the physician for a specific treatment because others recommended it and the physician does not find it suitable for a patient a discussion may arise where the patient insists on the desired treatment. If the physician chooses the lesser treatment because the patient insists, the patient may leave the office very satisfied, but with a less optimal treatment than the physician would have chosen otherwise. Patient satisfaction would come from the care provider satisfying all patient's demands (sick-leave, more expensive drugs, unnecessary investigations). This may however lead to worse outcomes for the patient and the community as a whole.
Author Response
|
Reviewer’s Comments |
Author’s Response |
|
|
Thank you. Down side these phenomena has been added in the literature review section. Highlighted in blue |
|
Thank you. We agree with this comment. We have added this point as a limitation of the study. |
|
|
Thank you for your value comments. We have discussed the downside of storytelling, patient empowerment and the dimension of satisfaction. We have taken this suggestions and updated a the literature review and limitation sections accordingly. Please see the text highlighted in blue in the Literature review section. |

Round 2
Reviewer 1 Report
Thanks to the authors for the consideration of recommendations and comments. It is still recommended to add the numerical results to abstract and conclusion sections.
Author Response
Thank you for your consideration and valuable input. As suggested, we have included key numerical in the abstract and throughout the discussion section.
Reviewer 2 Report
Thank you for addressing my comments.
Author Response
Thank you for your valuable input.
Reviewer 3 Report
The authors have adressed all my comments adequately and taken the time to further search the litterature in response to my comments
Author Response
Thank you for your suggestions.